# Polylactic Acid and Polybutylene Succinate Biopolymer Blends for Extrusion Processing: Dry Blending vs. Masterbatch Dilution

**DOI:** 10.3390/polym17233117

**Published:** 2025-11-24

**Authors:** Milad Azami, Atul Kumar Maurya, Ramaswamy Nagarajan, Amir Ameli

**Affiliations:** Department of Plastics Engineering, University of Massachusetts Lowell, 1 University Ave., Lowell, MA 01854, USA; milad_azami@student.uml.edu (M.A.); atul_maurya@uml.edu (A.K.M.); ramaswamy_nagarajan@uml.edu (R.N.)

**Keywords:** biopolymer, polylactic acid, extrusion, compounding, blend, mechanical properties

## Abstract

Environmental concerns about plastic waste have increased interest in biobased and biodegradable polymers such as polylactic acid (PLA) and polybutylene succinate (PBS). Blending PLA and PBS can provide a balanced performance, offsetting the PLA’s brittleness. PLA/PBS can be processed either via single-screw extrusion (SSE) or twin-screw extrusion compounding followed by SSE (TSSE). This study aims at a comprehensive investigation of these two processing routes and assesses their impact on the physical, morphological, and mechanical properties of PLA/PBS blends. The results indicate that while both routes produce blends with comparable overall performance, subtle differences exist in the degradation behavior of PLA and the morphology of the blends. The PLA molecular weight drop was more pronounced in TSSE (~18.7%) compared to SSE (~1.5%). In both processing routes, PBS exhibited sub-micrometer domains below 15 wt.% loading, beyond which a distinct sea–island morphology with larger PBA domains was observed. TSSE exhibited slightly finer PBS domains. However, these differences did not lead to significant mechanical performance or miscibility differences. For instance, with 15 wt.% PBS loading, the elongation at break was improved from 4.6% to 193% in SSE15 and 192% in TSSE15, with a 29% and 30% decrease in yield strength, respectively. This work suggests that the single-step SSE process can be used as a cost-effective and energy-saving approach in PLA/PBS blending without the need for pre-compounding.

## 1. Introduction

In recent years, the escalating sustainability concerns of petroleum-based plastic materials have led to a significant growth of attention to biobased polymers. Polylactic acid (PLA) is a promising plant-based biodegradable polymer with high-scale production feasibility and similar stiffness to commodity polymers [1]. However, the inherent brittleness of PLA, low melt strength, slow crystallization kinetics, and processing challenges have restricted its utilization in large-scale manufacturing [2,3,4,5]. Various efforts have been made to alter PLA’s properties by blending with secondary biopolymers such as lignin [6,7,8] through methods like solution mixing [6] and melt mixing [9].

Polybutylene succinate (PBS) is another biodegradable, biobased aliphatic polyester, which has been studied for blending with PLA, aiming to possibly increase the PLA’s toughness or alter its crystallinity [9,10,11,12]. Deng and Thomas compounded PLA and PBS using a counter-rotating mixer (Haake Rheomix OS) at 175 °C, then hot-pressed at 190 °C for subsequent characterization. They found that adding PBS in the range of 10–40 wt.% to PLA increases PLA’s elongation at break up to 270%. This improvement was mainly attributed to the finely dispersed morphology of the PBS phase [13]. In other work, Ostrowska. et al. blended PLA and PBS using a co-rotating twin-screw extruder. They observed that the elongation at the break in PLA/PBS blends was distinctly improved (over 250%) only when the PBS ratio is at 50 wt.% and above [14]. Moreover, Yokohara and Yamaguchi showed that the presence of PBS in PLA/PBS blends, even in the molten state, can accelerate the crystallization of PLA, due to the crystal-nucleating effect of impurities in PBS or molten PBS itself [9]. On the other hand, Wang et al. reported that PBS does not trigger crystallization or increase the crystallinity of the PLA matrix during the typical melt compounding process [15].

It is known that the mechanical and thermal properties of the final product are directly influenced by the quality of mixing when multiple materials are employed. A good mixing can result in enhanced material performance and thus more competitive end products [9,11,14]. Most studies on PLA/PBS blends have employed a melt-blending process. Melt blending is one of the most cost-effective and scalable methods to engineer the attributes of a primary polymer. The continuous melt processing machines are always associated with an extruder, since its basic role is to pump the melt through a shaping die [16]. The most frequently used extruders are single-screw extruders (SSE), which offer simplicity and high throughputs, compared to twin-screw extruders (TSE) [17,18].

When compared to TSEs, SSEs are commonly known to be less proficient mixers; despite this, their mixing performance (both with unmodified [19] and modified screw designs) has been the focus of numerous studies aimed at enhancing mixing efficiency for various traditional polymers [20,21,22]. Recent studies have investigated the design and assessment of mixing elements for SSEs. Wang et al. conducted a numerical simulation to evaluate the distributive mixing performance in a twin-flight SSE, employing various screw designs [23]. Pandey incorporated extensional mixing elements (EME) into SSE, enhancing the dispersive mixing capability [18] and found that hyperbolic contracting–diverging channels of EME promoted the polymer droplet breakups and hence helped in the better distribution of the secondary polymer into the primary phase, compared to only pin-type extruders for both SSE and TSE.

Prior to processing with a regular SSE, compounding using a TSE process is often favored in hybrid material systems, as TSE can provide superior dispersive mixing [24,25]. However, it entails increased energy consumption, an extra step in the manufacturing cycle, and additional time and effort. Fortunati et al. used a masterbatch preparation strategy to prepare a PLA/PBS/plasticizer masterbatch for better mixing in the film extrusion process [26]. Meng et al. [27] fabricated PLA-poly (butylene succinate-co-butylene adipate) (PBSA)-starch hybrid system in one-step and two-step extrusion processes. While processing technique and conditions strongly influence the degree of mixing, an additional consideration for biodegradable polymers is their sensitivity to thermal degradation [5]. Mysiukiewicz et al. studied the impact of processing parameters on the degradation of PLA through TSE. They showed that high-MFI, low-viscosity grades degrade less due to shorter residence times, with initial molecular weight having minimal impact [28]. Similar conclusions were reported by Aldhafeeri et al., wherein they investigated the effects of extruder type, screw configuration, screw speed, and feed rate on PLA degradation using quad-screw and twin-screw extruders and found that reducing residence time is the primary factor in minimizing process-induced degradation [29].

The TSE process of masterbatch making followed by SSE dilution is believed to provide superior dispersion due to the selectivity of screw profiles and higher shear stresses. Even though various reports on the fabrication and characterization of PLA/PBS blends are available, the literature lacks a systematic study on the assessment of dry blending against the masterbatch route. It is of great importance to explore if a direct SSE process with dry pre-hopper mixing of PLA and PBS would suffice and provide effective mixing of the biopolymer blends. This could result in significant energy, cost, and time savings toward a cleaner production of PLA/PBS bioplastics.

Therefore, in this study, we report an in-depth experimental exploration of the two primary melt blending processing methods for PLA/PBS bioplastic blends. In the first method, referred to as the SSE route, the virgin PLA and PBS pellets are pre-mixed at the desired compositions in a dry state right before feeding the hopper and then melt-processed and mixed using an SSE machine. The SSE machine was equipped with a screw having a pin-type profile at the metering section for better mixing [30]. The second method is the masterbatch route, which uses a TSE process to prepare the PLA/PBS masterbatch, followed by an SSE process to dilute the concentrate by the addition of virgin PLA to prepare the blends with the final desired compositions. This route was called TSSE.

For a comprehensive comparison, morphological, thermal, rheological, and mechanical characterizations were performed. Process-induced degradation of PLA was evaluated using gel permeation chromatography and Fourier transform infrared spectroscopy. Thermal properties such as crystallinity, cold crystallization temperature, and glass transition temperature (T_g_) were analyzed using differential scanning calorimetry. Dispersion and distribution of the PBS domains in PLA were examined using scanning electron microscopy and parallel plate rheometry. The mechanical performance of the blends was also assessed using a tensile test. The relative behavior and performance of the SSE and TSSE blends were contrasted and discussed in detail to draw conclusions about the efficiency of SSE against TSSE.

## 2. Materials and Methods

### 2.1. Materials

PLA (Ingeo™ Biopolymer, 6202D) with a density of 1.24 g/cm^3^ and a melt flow index (MFI) of 15–30 g/10 min (210 °C) was kindly provided by NatureWorks (Minnetonka, MN, USA). PBS (FZ78TM) with a density of 1.26 g/cm^3^ and an MFI of 22 g/10 min (190 °C) was purchased from Mitsubishi Chemicals (Don Hua Lor, Thailand). PLA and PBS pellets were dried at 60 °C for 8 h prior to any melt processing. 

### 2.2. Twin-Screw Extrusion Processing

For the TSSE route, the pellets were first compounded in a Leistritz twin-screw extruder (ZSE 18HP) (Nuremberg, Germany) with a diameter of 18 mm and a length-to-diameter (L/D) ratio of 40. The machine had a modular programmable screw, and Figure 1 shows the utilized screw program together with the set barrel temperature profile (160–195 °C) and the screw rotational speed (200 rpm). The kneading block elements of various staggering angles (30, 60 and 90°) were incorporated. The stagger angle of kneading blocks controls mixing intensity, shear, and conveying in twin-screw extrusion. Small angles like 30° promote gentle mixing and fast conveying, medium angles (e.g., 60°) give balanced mixing, and large or neutral angles (e.g., 90°) deliver high shear with reduced conveying for intensive melt work. Kneading block elements were incorporated in multiple mixing zones with a goal of maintaining a balance between a high level of PBS dispersion in PLA and a low level of process-induced degradation. The dried PLA and PBS pellets were introduced to the hopper through separate volumetric feeders (Brabender), with adjusted feed rates that provided a PLA/PBS ratio of 60/40 wt.%, resulting in a total output of 5.3 kg/h. This masterbatch was later diluted with a single-screw extruder, as detailed in the next section. Pristine PLA pellets were also extruded under the identical processing conditions as the control sample.

### 2.3. Single-Screw Extrusion Processing

For single-screw extrusion, a Collin extruder (E30P) (Maitenbeth, Bavaria, Germany), with a screw diameter of 30 mm and an L/D ratio of 25, was employed. As shown in Figure 2, the screw profile consisted of pin-type elements at the metering zone. Pin-type elements can promote the elongational flow and cut the melt inside the barrel multiple times to enhance the distributive mixing [31]. The screw speed was kept at 20 rpm, and the barrel temperature profile is given in Table 1.

As previously mentioned, the samples were prepared through two distinct processing routes. In the first route, referred to as SSE (single-screw extrusion), virgin PLA and virgin PBS pellets were dry-mixed during feeding according to the specifications in Table 2. For the TSSE (twin-screw followed by single-screw extrusion) route, the 40 wt.% PBS masterbatch was further processed by SSE and diluted with virgin PLA pellets to various concentrations, as outlined in Table 2. Subsequently, all extrudates underwent water cooling and pelletizing.

### 2.4. Characterizations

Gel permeation chromatography (Alliance 2695, Waters Corp., Milford, MA, USA) was conducted, using tetrahydrofuran (THF) as the mobile phase, together with polystyrene as the standard reference samples. A 40 mg sample of PLA was dissolved in 10 mL of THF by stirring at 55 °C for 24 h. The number-average and weight-average molecular weights (M_n_, M_w_) as well as the polydispersity index (PDI) of three sample repeats were obtained, and the average values are reported.

### 2.5. Fourier Transform Infrared Spectroscopy

Fourier transform infrared spectroscopy (FTIR) tests were conducted in attenuated total reflectance (ATR) mode. A sNicolet™ iS50 spectrometer from Thermo Fisher Scientific (Waltham, MA, USA) was used to conduct the FTIR spectroscopy of virgin PLA (VPLA) and extruded PLA/PBS blends. The analysis was conducted in the wavelength range of 500 to 4000 cm^−1^. To assess the level of process-induced degradation, the carbonyl index values of VPLA, SSE0, and TSSE0 were calculated using Equation (1):(1)$$Carbonyl\;Index\;\left(CI\right)=\frac{Absorption\;band\;area\;at\;1747\;cm^{-1}}{Absorption\;band\;area\;at\;1452\;cm^{-1}}$$
where the area peaked at 1747 cm^−1^ is associated with the C=O (carbonyl) stretching vibration, and 1452 cm^−1^ is the absorption band accounting for C–H bending vibration [14,32]. For this analysis, spectra were collected from three different samples of each composition, three times. Mean values and standard deviations of nine runs were calculated and reported. The area under the absorption bands was calculated using OMNIC (9.12.928) software.

### 2.6. Differential-Scanning Calorimetry

Differential-scanning calorimetry (DSC) was performed using TA Instruments (DSC 2500 series) (New Castle, Delaware, USA) with aluminum pans. The heat–cool–heat procedure was applied for 5–6 mg of extruded pellets under a nitrogen atmosphere. Samples were heated at a 10 °C/min rate, up to 200 °C, and were held for 5 min. The samples were then cooled to 25 °C with a cooling rate of 5 °C/min. The second heating cycle was performed with a rate of 10 °C/min up to 200 °C. The cold crystallization temperature (T_CC_) and the melting point (T_m_) were obtained from the second heating thermograms. The crystallinity of PLA (X_C_) was calculated using Equation (2):(2)$$X_{c}\left(\%\right)=\frac{∆H_{m}-∆H_{cc}}{{∆H}_{m}^{0}.w}$$
where w represents the weight fraction of PLA in the blend. ∆H_m_ and ∆H_CC_ refer to the melting enthalpy and cold crystallization enthalpy values of PLA, respectively, as measured from the DSC thermograms. The value of ∆H^0^_m_ denotes the enthalpy of fusion for 100% crystalline PLA which is reported as 93 J/g [33]. The crystallinity of virgin PBS (VPBS) was also found using the same equation where w = 1 and ∆H^0^_m_ = 110.3 J/g [12].

### 2.7. X-Ray Diffraction Analysis

X-ray diffraction measurements were conducted using a Rigaku X-ray diffractometer (Rigaku Corporation, Tokyo, Japan) utilizing a copper X-ray tube with a wavelength of 0.154 nm to evaluate the crystallinity of all the samples prepared using micro injection molding. The tests were conducted with a scan speed of 3° per minute and a step size of 0.01°. The samples were scanned over a 2θ range from 5° to 50°.

### 2.8. Scanning Electron Microscopy

The microstructure of the cryo-fractured samples was observed using a JEOL (JSM 7401F) electron microscope (Tokyo, Japan) at an acceleration voltage of 5 kV. The samples were gold sputter-coated for 180 s before observations.

### 2.9. Rheological Tests

The parallel plate rheological tests were carried out using ARES-G2 rotational rheometer from TA Instruments (New Castle, DE, USA), following the ASTM D4440 standard. Test specimens, disks with a diameter of 25 mm and a thickness of 2 mm, were prepared using a micro injection molding machine (Xplore IM 12) (Sittard, The Netherlands). All tests were performed at 190 °C. To eliminate sample oxidation effect, nitrogen gas was purged into the chamber throughout the tests. The tests were performed in frequency sweep mode within a range of 0.1–100 Hz. The strain amplitude was set at 10%, which was found to fall within the linear viscoelastic region.

### 2.10. Mechanical Tests

For mechanical tests, ASTM D638 type-V tensile specimens were prepared with a micro injection molding machine (Xplore IM 12) (Sittard, The Netherlands). The blends were tested (4 replicates) under tension using an Instron 5966 (Norwood, MA, USA) with a 10 kN load cell. The displacement rate was set at 10 mm/min (ASTM D638 standard).

For GPC, FT-IR, and rheology tests, 3 replicates were used, and for DSC and XRD tests, 2 replicates were used.

## 3. Results and Discussion

### 3.1. Process-Induced Degradation

To assess the impact of the two processing routes on the potential degradation of PLA, GPC, and ATR-FTIR tests were conducted on 100% PLA samples (VPLA, TSSE0, and SSE0), and zero-shear viscosity was determined from oscillatory viscosity curves, and the results are summarized in Table 3. The weight-average molecular weight (M_w_) showed a slight decrease from VPLA (179.6 kDa) to SSE0 (176.9 ± 4.6). However, this reduction was more significant in TSSE0 (146.1 ± 14). The polydispersity index (PDI) followed a similar decreasing trend. Whereas the number-average molecular weight (M_n_) remained relatively unchanged. A similar behavior has been previously reported for PLA during twin-screw extrusion [34], which could be attributed to a somewhat selective molecular cleavage through degradation mechanisms that preferentially affect longer chains, as they have a higher probability of defects, a large number of susceptible linkages, and experience a larger degree of entanglement [34].

Zero-shear rate viscosity, measured at low shear rates, is usually correlated with molecular weight and can serve as another indicator of degradation [28,29,35]. The zero-shear viscosity values, derived from the complex viscosity obtained through oscillatory rheometry, are reported as the average of three measurements in Table 3. VPLA exhibited a zero-shear viscosity of 591.3 ± 67 Pa.s, SSE0 showed a slight reduction in zero-shear viscosity (549 ± 84.2 Pa.s), and TSSE0 showed a more severe reduction (343 ± 19.3 Pa.s).

The carbonyl index can also be calculated to assess the degree of degradation of PLA, which undergoes melt processing [29]. There have been different methods suggested to carry out this analysis [36]. In this study, the carbonyl index was determined using the area under the curve method. Specifically, the area under the absorption band of carbonyl stretching (1747 cm^−1^) was calculated and normalized to the area under the curve of C-H vibration (1452 cm^−1^). As it is shown in Table 3, the CI from VPLA (4.6 ± 0.31) to SSE0 (4.9 ± 0.18) slightly increases, and increases more in TSSE0 (5.8 ± 0.31). The increase in the carbonyl index results from the thermomechanical degradation of PLA, resulting in chain scission and formation of new carbonyl groups. This process reduces the material’s overall molecular weight and alters the carbonyl absorption band intensity through the formation of anhydrides and carboxyl groups during melt processing [28,29,37].

These degradation assessments demonstrated that twin-screw extrusion (TSE) induces more pronounced polymer degradation compared to single-screw extrusion (SSE), despite having a shorter residence time. The higher shear stress levels in TSE lead to more significant molecular weight reduction and chemical changes, while the changes observed in SSE0 remain negligible. This suggests that the intense mechanical processing in TSE can cause more substantial structural alterations to the polymer during the masterbatch preparation for the TSSE route.

### 3.2. Thermal Behavior

Figure 3a,b present the DSC thermograms for the second heating cycle of VPLA, SSE0, TSSE0, VPBS, as well as the blends for both processing routes. Table 4 lists the glass transition temperature (T_g_), cold crystallization temperature (T_cc_), cold crystallization enthalpy (H_cc_), the heat of melting enthalpy (H_m_), and the melting temperature (T_m_). Overall, T_g_ changes only slightly. VPLA recorded a T_g_ of 60.5 °C, which was the highest among all the samples. SSE0 and TSSE0 reported a T_g_ of 59.9 °C and 59.8 °C. A slight decrease in the T_g_ of extruded neat PLA samples can be ascribed to the chain scission happening during the processing. Ramos-Hernández et al. [38] and Rasselet et al. [39] reported similar findings. Furthermore, with increasing PBS content, all the blends showed a mild decrease in T_g_, which was the lowest (58.2 °C) for the TSSE40 sample. It was presumed that the lower T_g_ of the VPBS (−33.8 °C) contributed to this decrement in the blends [40].

As seen in Figure 3, VPLA exhibits a significantly higher cold crystallization temperature (120 °C) than VPBS (100 °C). This difference is due to the PLA’s slower crystallization kinetics compared to that of PBS [11]. Further extrusion of the PLA (SSE0 and TSSE0) reduces the T_cc_ to 110 °C. Adding PBS up to 20 wt.% further reduces T_cc_ to 104 and 102 °C for the SSE and TSSE routes, respectively. However, at 40 wt.% PBS, the T_cc_ slightly increases to 105 and 104 °C in SSE and TSSE routes, respectively. The melting temperature (T_m_) of PLA remains relatively stable around 167 °C for all the samples in both routes, indicating that the presence of PBS and processing route have a minimal impact on the melting behavior of PLA. The normalized enthalpy of PLA (H_m_) was obtained by dividing the melt enthalpy value of PLA by its weight fraction in each blend. Both cold crystallization enthalpy (H_cc_) and normalized Hm of PLA showed a consistent decreasing trend with PBS addition in both SSE and TSSE routes. These findings indicate that the crystallization behavior of PLA in the blends is primarily influenced by the PBS content, with a minor impact from the processing route. For the degree of crystallinity, VPLA exhibited a small crystallinity value of 0.1%. However, in the extruded SSE0 and TSSE0 samples, the crystallinity increased to 2.3% and 3.3%, respectively. This mild increase can be attributed to the lower molecular weights of SSE0 and TSSE0, as indicated by GPC results, where the polymer chains gained more mobility and thus more easily arranged into crystalline domains. Moreover, Ostrowska [14] suggested that chemical species resulting from the degradation of PLA, such as oligomers, can act as additional nucleating agents and thus increase the crystallinity. Notably, the higher crystallinity of TSSE0 compared to SSE0 aligns well with the greater average molecular weight loss observed in TSSE0. VPBS exhibited a cold crystallization peak just before its melting endotherm. The onset temperature and enthalpy of cold crystallization were 95 °C and 6 J/g, respectively. This peak was immediately followed by a melting endotherm with an enthalpy of 60 J/g and an onset temperature of 107 °C, resulting in a crystallinity of 48.8%.

The cold crystallization peak temperature of PLA shifted down upon the addition of 5 wt.% PBS in both SSE (from 110 °C to 104 °C) and TSSE (from 110 °C to 106 °C) cases. Further addition of PBS in both groups resulted in only additional changes of 1–3 °C. It was observed that this exothermic peak fully or partially overlapped with the endothermic melting peaks of PBS, as they occur within a similar temperature range. This overlap complicates the precise calculation of PLA crystallinity within the blends, in particular as higher PBS loadings, as also discussed in [13,14,41,42]. Therefore, the crystallinity values provided in Table 4 are only estimates and should be treated with caution.

There are several controversial studies regarding the effect of PBS on the crystallinity of PLA. Ji et al. [43] did not observe any crystallization of neat PLA and PLA blended with 20 wt.% PBS, whereas neat PBS underwent crystallization. Yokohara et al. [9] studied the blends containing PBS component ranging from 5 to 20 wt.% and reported that PBS droplets or impurities in the PBS during the cooling of the PLA/PBS blends act as the nucleating sites for PLA. Wang et al. [15] reported that PBS does not trigger crystallization or increase the crystallinity of the PLA matrix, and PLA remained amorphous in all their samples having 10–40 wt.% PBS. It appears that the effect of PBS on PLA’s crystallinity is not consistent and depends on several other factors, such as blend composition, miscibility, processing methods (which dictate the blend morphology), and variations in polymer molecular structures, such as different D content in PLA and various additives.

### 3.3. XRD Studies

To further analyze crystallinity, XRD measurements were carried out. The XRD patterns of VPLA, VPBS, and extruded blends of both SSE and TSSE routes are presented in Figure 4. The most notable feature of the XRD pattern for VPLA was a broad halo lacking any sharp peak. This broad peak, which is centered around the 2θ angle of 16°, indicates that VPLA remained amorphous during the micro-injection molding. In contrast, the XRD pattern of VPBS shows distinct crystalline peaks at the 2θ angles of 19.5°, 22.5°, and 28.8°, associated with the (020), (110), and (111) crystal planes of α-form PBS, respectively [15,44]. For the blends, upon addition of 5 wt.% PBS, the characteristic crystalline peak of PLA around 2θ = 16.5° appears; the intensity of this peak persists up to 15 wt.% PBS in both routes, indicating the creation of crystal domains in the PLA in the presence of PBS. It can be said that, in this range of PBS loading, small PBS domains form nucleating sites in the amorphous domain of PLA and improve the chain flexibility of PLA. However, with further increases in PBS to 20 wt.% and 40 wt.%, the peak 2θ = 16.5° of PLA attenuated, and the crystalline peaks of PBS became more pronounced. This is likely due to phase separation, which limited further PBS from entering the PLA-rich phase [43]. Similar findings have been previously reported in XRD studies on other polymer–polymer blends [45,46]. The XRD measurements and DSC tests were conducted on the same microinjection-molded samples. The crystallinity trends obtained from XRD were generally consistent with those from the first DSC heating scans (Table 4). This agreement holds for PBS contents up to 15 wt.% in both processing routes. Comparing the XRD and DSC data of the two processing routes up to 15% PBS, the TSSE samples consistently exhibited slightly higher crystallinity than the corresponding SSE samples. This is reflected in the sharper XRD peaks (Figure 4b) and the higher first-scan crystallinity values in Table 4. The additional processing step in TSSE further reduces the molecular weight of PLA, as discussed in Section 3.1, which likely increases chain mobility and contributes to the higher degree of crystallinity.

### 3.4. Chemical Structure and Blend Morphology

Figure 5a–d provide the ATR-FTIR spectra of the VPLA and VPBS and PLA/PBS blends processed via both routes. VPLA and VPBS exhibited absorption bands corresponding to C=O (carbonyl) stretching due to having ester groups at 1747 cm^−1^ and 1712 cm^−1^, respectively [14]. In the spectra of blends from both routes, compared to VPLA and VPBS, no new peaks emerged or disappeared, indicating that no chemical reactions occurred between PLA and PBS [47]. Both the SSE and TSSE series showed characteristic peaks of PLA and PBS, with peak intensities changing accordingly as the PBS content varied. In the blends with PBS content up to 10 wt.%, the C=O stretching vibration absorption band was a single modal peak around 1747 cm^−1^ with a slight broadening with an increase in the PBS content. The 1712 cm^−1^ C=O stretching peak of PBS was not clearly visible up to a PBS loading of 10 wt.% in both SSE and TSSE samples. This could imply some level of miscibility between PLA and PBS at these ratios. As the PBS content exceeded 10 wt.%, the C=O stretching vibration absorption band transformed from a singlet to a broader doublet in the blends, and the peak near 1712 cm^−1^ became more prominent, eventually forming a single peak in VPBS. This trend was consistent across both TSSE and SSE processing routes with no significant identifiable differences (Figure 5c,d). The doublet peaks for C=O in the blends containing 20 and 40 wt.% PBS, compounded through both SSE and TSSE, indicate the presence of two separate phases of PLA and PBS at these compositions.

SEM micrographs of the blends and neat PLA are presented in Figure 6. In the case of neat PLA samples (Figure 6a,b), a relatively smooth surface with a characteristic of a brittle structure was observed. Upon the addition of PBS, up to 10 wt.% in both SSE and TSSE routes, PBS domains were scarcely distinguishable from the PLA phase. However, at 15 wt.% PBS (Figure 6g,h), in both SSE and TSSE routes, small discrete droplets of PBS appeared. By increasing PBS content to 20 wt.% (Figure 6i,j), a distinct interface between the two blend components emerged, where the PBS component exhibited slightly increased domain size. Despite the thermodynamic incompatibility of PLA and PBS [48], PBS domains remained in the submicron range in SSE20 and TSSE20. At 40 wt.% PBS, due to the coalescence phenomena, the domain size of PBS increased [49], reaching a size of several micrometers and increasing the size non-uniformity. Additionally, the spherical shape changed into slightly elongated droplets [50].

In a well-mixed blending process, the minor phase should maintain a consistent size (dispersive mixing) and equal spatial distribution in the major phase (distributive mixing) [18]. This was observed at PBS loadings of below 20 wt.% in both SSE and TSSE samples, where distinguishable PBS domains are uniformly spaced with minimal variation in size. This uniform distribution indicates effective distributive mixing. It is evident that there is no substantial difference in the morphology of TSSE and SSE samples at the same PBS concentrations. These findings indicate that both processing routes resulted in a similar state of dispersion of PBS within the PLA matrix.

### 3.5. Rheological Analysis of Blend Miscibility

Rheological measurements were conducted on all blends as well as the virgin polymers. VPLA exhibited a higher complex viscosity with a more extended Newtonian regime, whereas VPBS showed a much shorter Newtonian plateau compared to the PLA samples. Additionally, VPBS displayed slightly greater melt elasticity than VPLA, as it had a greater storage modulus in lower frequencies.

Figure 7 shows the Cole–Cole plots of the blends. A Cole–Cole plot is a useful tool to examine the multi-phase polymeric systems [51]. In this approach, a graph is plotted to display the relationship between the imaginary and real parts (*η*″ and *η*′, respectively) of the complex viscosity. The values of *η*″ and *η*′ are calculated from the following equations:(3)$$\eta^{\prime }=G^{\prime \prime }/\omega$$(4)$$\eta^{\prime \prime }=G^{\prime }/\omega$$
where ω is the frequency and *G*″, *G*′ are loss and storage moduli, respectively.

In both routes, for neat PLA at 5 wt.% and 10 wt.% PBS concentrations, a smooth semi-circular single arc shape was obtained. This indicates a single relaxation time in those samples and therefore shows the homogeneity of the blend [52]. When increasing PBS to 15 wt.% and 20 wt.% in both routes, the curves deviate from the single arc shape and change into a double arc shape. Furthermore, in the curves of 40 wt.% PBS in both routes, an extension emerges on the right side of the single arc. In general, these deviations from a single arc shape are an indication of immiscibility and phase separation, which result in the presence of two distinct relaxation times [52]. It is noted that this observation relates to the characteristics exhibited by the material in its molten state at 190 °C. However, it is interesting to note that the Cole–Cole findings are consistent with the SEM morphological analysis, which reveals the state of the blend after cooling and at room temperature. All in all, FTIR, SEM, and rheological analyses unveil that at low loadings of PBS, dry blending can disperse PBS into PLA to the same level as TSSE.

### 3.6. Mechanical Performance

The mechanical properties of the blends obtained from the two routes were evaluated using tensile tests, and the results are shown in Figure 8 and Figure 9. The representative tensile stress–strain curves of PLA, PBS, and PLA/PBS blends are given in Figure 8. As expected, the stress–strain curve of the VPLA, Figure 8a, demonstrated a brittle behavior without necking and the highest yield stress (YS) of 58.2 ± 1.3 MPa and Young’s modulus of (2814 ± 108 MPa), reaching to elongation at break (EAB) of about 3.5%. VPBS, as the toughening phase, recorded the highest EAB (322 ± 35%) and lowest YS (13.7 ± 0.1 MPa) and modulus (614.2 ± 11.8) among all the samples. Both SSE0 (52.3 ± 1.6 MPa) and TSSE0 (50.6 ± 1.3 MPa) exhibited reductions in tensile strength of 8.8% and 10.1%, respectively, compared to VPLA, along with a more pronounced necking region in their stress–strain curves. This mechanical behavior is attributed to the decreased molecular weight of PLA during extrusion, where shorter molecular chains facilitate disentanglement, resulting in increased elongation but reduced strength. In contrast, VPBS showed the lowest ultimate tensile strength at 32.4 ± 0.9 MPa.

The GPC results showed that TSSE0 underwent a greater reduction in molecular weight (18.4% for TSSE0 vs. 1.5% for SSE0). This did not, however, directly correlate with the drop in the tensile strength. One reason for this could be the slightly higher crystallinity of TSSE0 compared to SSE0, which may be related to its lower PDI, promoting more organized molecular chain conformations. These structural features can help compensate for the mechanical strength loss typically associated with molecular weight reduction. It should also be noted that the decline in mechanical properties is not necessarily linear with the drop in molecular weight. For instance, Tomás Ramos-Hernández et al. [38] reported a decrease of ~14% in tensile strength after three processing cycles. Fernandez et al. [53] reported a 12.7% reduction in tensile strength of PLA after two cycles of processing despite a 37.7% decrease in molecular weight. Pillin et al. found that PLA’s property loss was not proportional to its molecular weight reduction: a 10.4% M_w_ drop in the first cycle led to only a 4.6% decrease in strength, and even after a 30.4% M_w_ drop over two cycles, the strength declined by just 12.3% [54]. These findings suggest that while tensile strength decreases after multiple processing cycles of PLA, the relationship with the molecular weight reduction is not strongly linear [55], and the overall mechanical performance cannot be solely associated with the molecular weight loss.

With respect to the PLA/PBS blends, the YS consistently declined with increasing PBS content in both processing routes. In the SSE route, the YS dropped from 52.3 ± 1.6 MPa at SSE0 to 30.5 ± 1.5 MPa at SSE40. The TSSE route showed a similar decline, with the YS decreasing from 50.6 ± 1.3 MPa at TSSE0 to 31.3 ± 0.8 MPa at TSSE40. Several studies [14,56] indicate that incorporating PBS into PLA results in a continuous reduction in tensile strength. Bhatia et al. observed a pattern close to the rule of mixtures, a nearly linear interpolation between the tensile strengths of pure PLA and pure PBS in PBS contents below 20 wt.% [57]. The drop in the tensile strength arises primarily from two factors: PBS possesses lower ultimate tensile strength (32.4 ± 0.9 MPa) compared to VPLA (67.2 ± 1.9 MPa), and poor interfacial interaction between PBS and PLA phases results in ineffective stress transfer due to the immiscibility in the absence of compatibilizers.

The SSE0 and TSSE0 samples exhibited tensile moduli of 2496 ± 147 and 2568 ± 112 MPa, lower than the VPLA (2759 ± 149 MPa). With the incorporation of the PBS (608 ± 17.3 MPa), the moduli of the blends decreased further due to the more compliant nature of PBS. Similar findings were reported by Chang et al. [58] and Qiu et al. [55] for 20 and 10 wt.% PBS/PLA blends. Both the blends reported a decrease in the tensile modulus of all the blends with the increasing PBS content. Like blends extruded from the SSE route, all the blends compounded from the TSSE route followed a very similar decreasing trend for the tensile modulus with increasing PBS wt.%. At 15 wt.% PBS, both SSE and TSSE blends showed improved ductility compared to neat PLA, with EAB increasing by more than 40 times for both, with a modest drop in modulus and strength.

The consistent change in modulus with increasing PBS content is attributed to the inherently lower stiffness of PBS compared to PLA. However, at low loadings (5–15 wt.% PBS), the modulus decreased slowly, due to the partial miscibility and ultrafine PBS domain size, aligning with the morphological observations in SEM micrographs and Cole–Cole plots of rheological analysis. However, at 20 and 40 wt.% PBS, the reduction became more pronounced, indicating the dominance of the softer PBS phase in the blend’s mechanical response and phase separation. The trend of elongation at break with PBS content also changed beyond 15 wt.% PBS loading; just a small addition of PBS (5 wt.%) in both routes increased the ductility significantly (127.2 ± 15.3% for SSE5 and 128.4 ± 4.9% for TSSE5) and continued to rise until 15 wt.% PBS, reaching 193.3 ± 25.6% (SSE15) and 192.2 ± 11.8% (TSSE15). However, beyond 15 wt.% PBS, elongation at break did not show a significant increase. This suggests a morphological shift in the blends beyond 15 wt.% PBS, where further improvements in ductility are limited, most likely due to poor interfacial adhesion between the PLA and large PBS domains.

Finally, an Analysis of Variance (ANOVA) (Figure 10) was conducted and revealed that the processing route (Factor A: TSSE vs. SSE) had no statistically significant effect on the tensile modulus, yield strength, and elongation at break, at a significance level of 0.05. Obviously, PBS loading level (Factor B) exhibited a very significant effect on all these responses. It can thus be concluded that the tensile mechanical performance of the blends made via SSE and TSSE is statistically the same with 95% confidence.

## 4. Conclusions

In this study, PLA and PBS were blended using two processing routes: dry blending with single-screw extrusion (SSE) and masterbatch twin-screw extrusion followed by dilution using single-screw extrusion (TSSE). Neat PLA showed greater molecular weight reduction in the TSSE case, indicating greater thermal degradation. However, this did not result in any significant differences in the mechanical performance of TSSE samples versus SSE counterparts. The rheological and morphological analyses revealed some partial miscibility and/or submicron PBS domain size in the PLA/PBS blends when PBS content was less than 15 wt.%. At high PBS loadings of 20 to 40 wt.%, phase separation occurred, resulting in a sea–island morphology with distinct large PBS domains. In this range, the reduction in stiffness became more pronounced, while the improvements in EAB were stopped. In particular, the mechanical testing showed that up to 15 wt.% PBS, elongation at break significantly increased, reaching up to about 195%, compared to 4.5% for neat, processed PLA, with a mild decline in strength (29.7% in SSE15 and 30.1% in TSSE15) and modulus (17.4% in SSE15 and 14.5% in TSSE15).

More importantly, the results confirmed that the choice between SSE and TSSE had no significant effect on the PLA/PBS blend’s tensile properties, thus suggesting that a single-step extrusion process with an appropriate screw design is adequate for blending shear-sensitive PLA and PBS biopolymers. The choice of SSE over TSSE offers a simpler, more cost-effective, and energy-efficient option for extrusion processing.

## Figures and Tables

**Figure 1 polymers-17-03117-f001:**
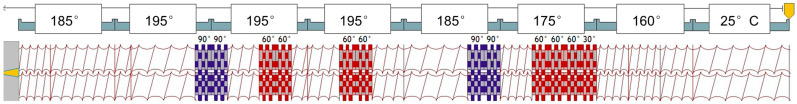
The utilized screw program and barrel temperature profile during the twin screw compounding of PLA/PBS:60/40 masterbatch. Red elements (30° and 60°) indicate forward-conveying components, while the purple (90°) elements represent natural-conveying components The numbers from right to left show the barrel temperature profile from feeding zone to the die exit.

**Figure 2 polymers-17-03117-f002:**
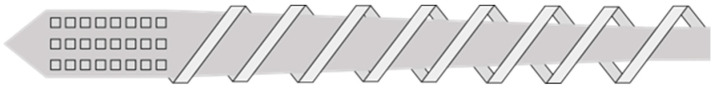
Screw design of the single screw extruder for the processing of PLA/PBS blends.

**Figure 3 polymers-17-03117-f003:**
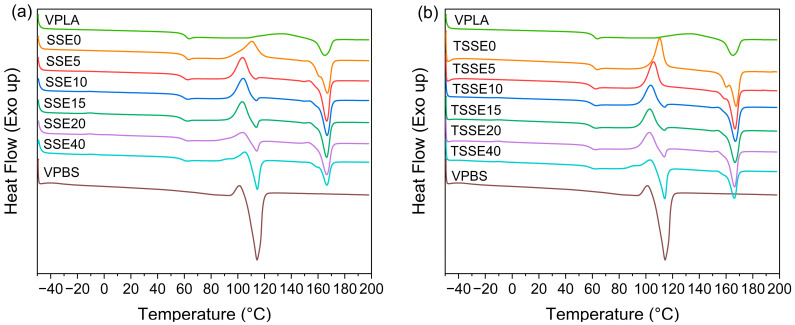
Second heating thermograms of virgin and extruded PLA, virgin PBS, and all the blends via (**a**) SSE route and (**b**) TSSE route.

**Figure 4 polymers-17-03117-f004:**
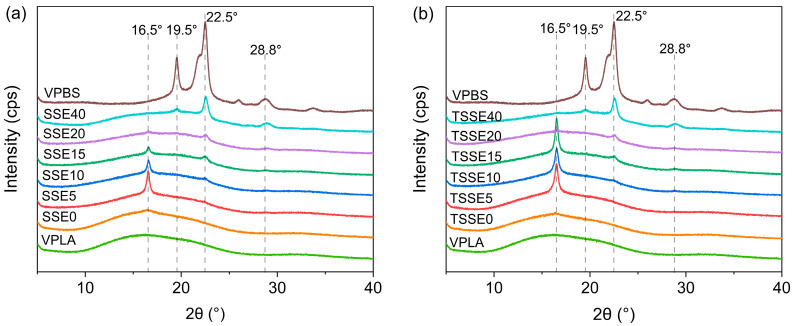
XRD patterns of VPBS, VPLA, and PLA/PBS blends for (**a**) SSE route and (**b**) TSSE route.

**Figure 5 polymers-17-03117-f005:**
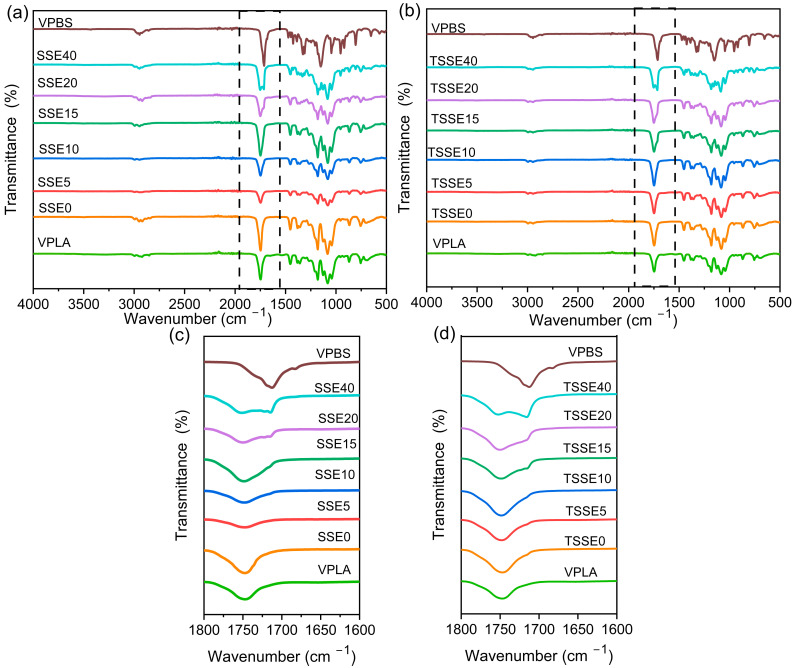
FTIR spectra of VPLA, VPBS, and PLA/PBS blends for (**a**,**c**) SSE route and (**b**,**d**) TSSE route.

**Figure 6 polymers-17-03117-f006:**
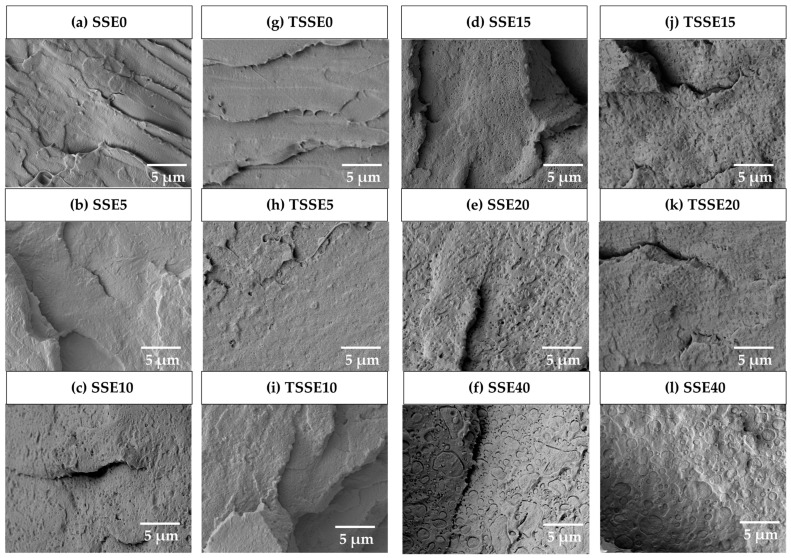
SEM micrographs of the processed samples of SSE route (**a**–**f**) and TSSE route (**g**–**l**).

**Figure 7 polymers-17-03117-f007:**
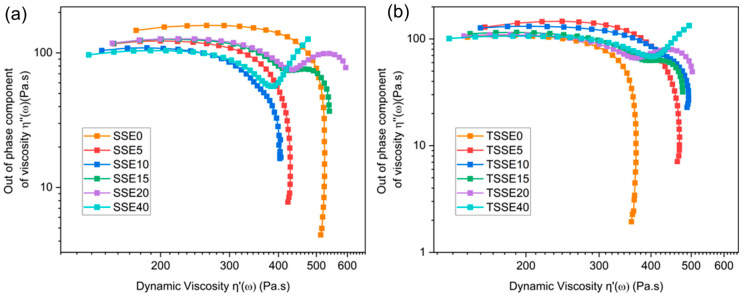
Cole–Cole plots of PLA/PBS blend processes through SSE (**a**) and TSSE (**b**) routes.

**Figure 8 polymers-17-03117-f008:**
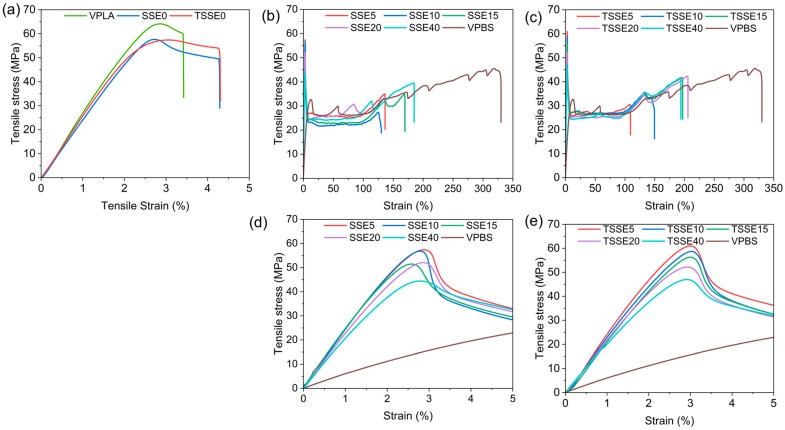
Representative tensile stress vs. strain curves of neat PLA samples (**a**), SSE routes (**b**), TSSE routes (**c**), SSE curve insets (**d**), and TSSE curve insets (**e**).

**Figure 9 polymers-17-03117-f009:**
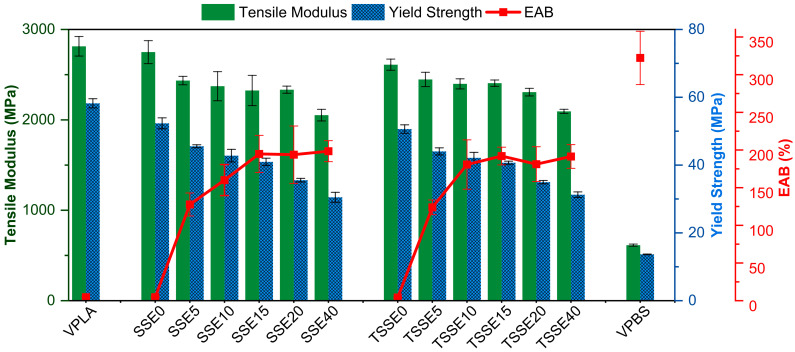
Tensile modulus, yield strength (YS), and elongation at break of VPLA, VPBS, and PLA/PBS blends processed through SSE and TSSE routes.

**Figure 10 polymers-17-03117-f010:**
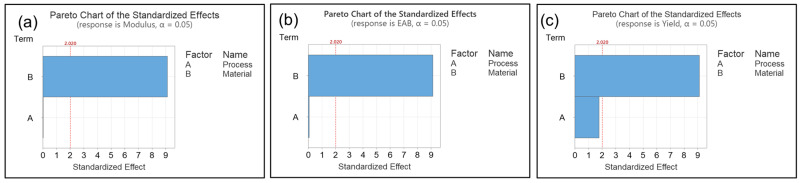
Pareto charts for the ANOVA analysis of modulus (**a**), elongation at break (EAB) (**b**), and yield strength (**c**), as a function of processing route (Factor A) and material composition (Factor B).

**Table 1 polymers-17-03117-t001:** Barrel temperature profile for single-screw extrusion trials.

Zone 1 (°C)	Zone 2 (°C)	Zone 3 (°C)	Zone 4 (°C)	Die (°C)
180	195	195	195	190

**Table 2 polymers-17-03117-t002:** PLA/PBS blend compositions prepared using two different routes: SSE and TSSE. The numbers following the letters in sample labels express the PBS weight percentage.

SSE Route	TSSE Route	PBS Composition (wt.%)
SSE0	TSSE0	0
SSE5	TSSE5	5
SSE10	TSSE10	10
SSE15	TSSE15	15
SSE20	TSSE20	20
SSE40	TSSE40	40

**Table 3 polymers-17-03117-t003:** Molecular weight (M_n_, M_w_), polydispersity index (PDI), zero-shear viscosity, and carbonyl index (CI) of virgin and processed PLA samples.

Sample	$\mathrm{Weight}\;\mathrm{Average}\;\mathrm{Molecular}\;\mathrm{Weight},\;M_{w}$ (kDa)	$\mathrm{Number}\;\mathrm{Average}\;\mathrm{Molecular}\;\mathrm{Weight},\;M_{n}$ (kDa)	Polydispersity, PDI	$\mathrm{Zero}\;\mathrm{Shear}\;\mathrm{Rate}\;\mathrm{Viscosity},\;\eta_{0}$ (Pa.s)	Carbonyl Index, CI
VPLA	179.6 ± 4.9	82.5 ± 4.9	2.19 ± 0.17	591 ± 67	4.6 ± 0.3
SSE0	176.9 ± 4.6	81.9 ± 4.1	2.17 ± 0.13	549 ± 84	4.9 ± 0.2
TSSE0	146.1 ± 14	82.2 ± 1.6	1.93 ± 0.21	343 ± 19	5.8 ± 0.3

**Table 4 polymers-17-03117-t004:** Glass transition (Tg), cold crystallization (Tcc), melting temperature (Tm), and normalized melt enthalpy values obtained from the second heating, and crystallinity values obtained from both the first and second heating scans.

Processing-PBS (wt.%)	T_g_ (°C)	T_cc_ (°C)	H_cc_ (J/g)	T_m_ (°C)	Normalized H_m_ (J/g)	X_c_ (%) First Scan	X_c_ (%) Second Scan
VPLA	60	120	17.4	165	17.5	0.7	0.1
SSE/TSSE0	59.8/60.3	110/110	28.6/27.9	167/167	30.7/30.9	5.3/7.3	2.3/3.3
SSE/TSSE5	58.0/58.7	104/106	22.3/27.6	167/167	30.3/30.9	10.9/11.2	9.1/3.7
SSE/TSSE10	58.4/58.7	104/103	22/21.3	167/167	27.0/28	10.3/13.3	6.0/8.1
SSE/TSSE15	58.2/58.7	104/102	18.5/20.1	166/166	26.3/27.1	10.8/18.2	9.8/9
SSE/TSSE20	58.7/58.7	104/102	10.3/17.7	166/166	25.2/25.5	16.4/14.6	20.0/10.5
SSE/TSSE40	58.2/58.2	105/104	9.1/11.5	166/166	17.9/19	9.0/17	15.7/13.4
VPBS	−33.8	100	5.57	114	59.4	49.7	48.8

## Data Availability

The original contributions presented in this study are included in the article. Further inquiries can be directed to the corresponding author.

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
