# Peer review of "Polylactic Acid and Polybutylene Succinate Biopolymer Blends for Extrusion Processing: Dry Blending vs. Masterbatch Dilution"

_polymers, 2025, doi:10.3390/polym17233117_

Round 1
Reviewer 1 Report
Comments and Suggestions for Authors
In recent years, growing environmental concerns related to the accumulation of plastic waste have prompted researchers and industry to seek more sustainable materials. Among them, biopolymers such as polylactic acid (PLA) and polybutylene succinate (PBS) have gained particular importance due to their biodegradability and origin from renewable resources.
Blending PLA with PBS enables the development of materials with more balanced mechanical properties, especially by reducing the brittleness of PLA. PLA/PBS blends can be processed either through single-screw extrusion (SSE) or twin-screw extrusion followed by dilution using SSE (TSSE). The aim of this study was to compare both processing routes and evaluate their influence on the physical and mechanical properties of the resulting blends.
The obtained results showed that both techniques led to comparable outcomes in terms of morphology, thermal characteristics, and mechanical performance. For example, at 15 wt.% PBS content, the elongation at break increased from 4.6% to approximately 193%, accompanied by a simultaneous decrease in yield strength by around 30%. This indicates that a single-step SSE process may represent a simpler, more cost-effective, and energy-efficient approach for industrial blending of PLA and PBS without the need for prior masterbatch preparation.
I have several comments, suggestions, and questions:
- I suggest expanding the literature review and discussion by including recent publications on the effect of extrusion method on the processing and selected properties of biopolymers.
- It is recommended to use SI units consistently.
- Were both series of samples (SSE and TSSE) prepared under identical cooling and pelletizing conditions?
- On what basis were the blend compositions selected for the experiments?
- If possible, please improve the quality of Figure 6.
- Were the Rabinowitsch and Bagley corrections applied during viscosity measurements and analysis?
- Figure 9 should be improved for better clarity and readability.
- Were the rheological tests performed in accordance with a specific standard?
- How many samples were tested for each type of experiment, and how was statistical significance ensured?
- Was any design of experiments (DOE) methodology applied?
- Do you think statistical data analysis using the ANOVA method should have been employed?
- How was the degree of homogenization and mixing of both phases evaluated?
- It would be valuable to add a short note describing the rheological differences between PLA and PBS.
Reviewer 2 Report
Comments and Suggestions for Authors
Abstract
- It states that 'overall, both processing routes resulted in similar morphological, thermal, and mechanical performance.'
It does not mention that PLA degradation is higher in TSSE (~18.4% vs 1.5% in SSE).
It does not discuss possible subtle differences in PBS dispersion or domain size, which could affect mechanical and thermal properties.
The sentence suggests complete equivalence, which is somewhat exaggerated.
- It states that with 15 wt.% PBS, EAB increased to 193% (SSE15) and 192% (TSSE15), while strength dropped by ~29–30%.
The values seem consistent with Section 3.6, but the Abstract does not explain why EAB increases so much, nor its relation to morphology and interfacial adhesion.
- The Abstract does not discuss:
The PBS range where partial miscibility occurs (<15 wt.%) and its effect on properties.
Phase morphology ('sea-island') and its influence on modulus and strength.
Introduction:
- "Polylactic acid (PLA) is a promising plant-based biodegradable polymer with high-scale production feasibility and similar stiffness to commodity polymers [1]."
Although PLA is a biodegradable polymer of plant origin, the sentence could be more precise by stating that its large-scale production is feasible but comes with challenges, such as high costs and technical limitations in large-scale manufacturing.
-"However, the inherent brittleness of PLA, low melt strength, and slow crystallization kinetics have restricted its utilization in large-scale manufacturing [2,3]."
The “low melt resistance” of PLA may be a vague description. Melt resistance is more likely related to its viscosity and rheological behavior rather than to “strength” itself.
-"Various efforts have been made to enhance PLA properties by introducing secondary polymer components [4–6] through methods like solution mixing [4] and melt mixing [7]."
The introduction of “secondary polymer components” should be better explained. The sentence may cause ambiguity regarding the nature of these components (e.g., copolymers, plasticizers, or other polymers).
A suggestion could be: “Various efforts have been made to improve the properties of PLA through the introduction of additional polymer components, such as copolymers or plasticizers, using methods such as solution blending [4] and melt blending [7].”
- Where the limitations of PLA (fragility, toughness, thermal properties, and processing) are discussed, I would suggest including the article DOI: 10.1002/pc.27012. This study addresses the mechanical and thermal limitations of PLA, including fragility and low toughness, and also discusses how the addition of graphene nanoparticles can improve these properties. I believe that including this reference would complement the section on the limitations of PLA mentioned in the manuscript.
- "Polybutylene succinate (PBS) is another biodegradable, biobased aliphatic polyester, which has been studied for blending with PLA, aiming to increase the PLA’s toughness and crystallinity [7–10]."
The relationship between 'crystallinity' and 'toughness' of PLA needs further clarification. An increase in crystallinity may not necessarily improve toughness, depending on the conditions.
“PBS, another biodegradable aliphatic polyester of plant origin, has been studied for blending with PLA to improve its toughness and possibly its crystallinity [7-10].” By using 'possibly,' the sentence becomes more precise.
- "Deng and Thomas found that adding PBS in the range of 10-40 wt.% to PLA increases PLA’s elongation at break up to 270%."
It would also be interesting to include a remark on the experimental conditions under which this improvement was observed (temperature, processing type, etc.).
- "Ostrowska. et al. observed that the elongation at the break in PLA/PBS blends was distinctly improved (over 250%) only when PBS ratio is at 50 wt.% and above [12]."
The sentence suggests that elongation improves significantly only when the PBS ratio is above 50%, but this needs further precision in terms of the experimental context.
- "Polymer breakups and hence helped in the better distribution of the secondary polymer into the primary phase, compared to only pin-type extruders for both SSE and TSE."
The sentence seems overly vague when describing the 'breakage' of the polymer. It could be more specific about the physical phenomenon, such as 'fracture' or 'failure' of the material.
- "Prior to processing with a regular SSE, compounding using a TSE process is often favored in hybrid material systems, as TSE can provide superior dispersive mixing and precise dosing control [22,23]."
The sentence assumes that the TSE provides 'precise' dosage control without explaining how this is achieved. The accuracy of the dosage depends on the extruder setup and the feeding strategy.
- "Even though various reports on the fabrication and characterization of PLA/PBS blends are available, the literature lacks a systematic study on the assessment of direct dry blending against the masterbatch route."
The use of “direct dry blending” seems confusing. “Direct dry blending” is not a widely accepted term in the scientific literature, and it is more common to refer to it simply as “dry blending” or “dry mixing” as a general description.
- "For a comprehensive comparison, various characterizations were conducted."
The sentence is vague. What specific characteristics were analyzed?
Materials and Methods
- "g/cm3”--> it should be g/cm³. Review the entire document.
- Sometimes it is written as ºC and other times as °C, when the correct form is “°C”. Review the entire text.
- The same issue occurs with the symbol º, related to degrees. The correct form is “°C”. Review the entire text.
- Twin-Screw Extrusion (TSE):
Explain in more detail the effects of these offset angles and how they impact the dispersion of PBS and the degradation of PLA. If possible, cite relevant literature that has already addressed these aspects.
- Single-Screw Extrusion (SSE):
The description of the Collin E30P extruder and the explanation of the pin-type screw profile are well described. However, the use of the pin-type could be further explored, as these elements promote elongational flow and may have significant implications for the efficiency of the mixing process.
The screw speed (20 rpm) is specified, but it might be helpful to explain why this particular speed was chosen (it could be a pre-established experimental condition or a reference from previous studies).
It would be interesting to compare the efficiency of SSE with TSE not only in terms of mixing rate but also in terms of thermal or chemical effects during processing, particularly regarding the impact on PLA degradation.
- Line 190 – “Scanning electron microscopy” is not in italics and does not have the same formatting. Standardize it.
Results and Discussion
- Section 3 presents the titles without numbering, only starting with 3.4… Review.
- Pa.S should be Pa·s for viscosity.
- Temperatures should have a space before °C. For example: 60 °C.
- “The increase in the carbonyl index results from the thermomechanical degradation of PLA, leading to polymer chain scission and the formation of new carbonyl groups.”
It can be more concise: “The increase in carbonyl index indicates thermomechanical degradation of PLA, resulting in chain scission and new carbonyl groups."
- Terms like “in both SSE and TSSE routes” appear frequently; consider alternating with “for both routes” to avoid repetition.
- Some long sentences could be divided for better readability, especially in the DSC and XRD paragraphs.
- Just a note: “severer reduction” → should be “more severe reduction”
- wight average molecular weight → correct to weight average molecular weight.
- nucleating cites → correct to nucleating sites.
- Standardize the use of the hyphen: “cold-crystallization temperature” and “cold crystallization enthalpy” → choose one consistent style.
- Spacing in numbers with ±: “146.1±14” → ideally “146.1 ± 14”.
- Be careful with “VPBS (-33.8 °C, line 261)” vs “VPBS (-29 °C, Table 4)”. Confirm which value is correct, as there is an inconsistency between the table and the text.
3.4. Chemical structure and blend morphology
- Line 360 – there is a reference to “Figure 4 (c, d)” — but the previous section ends at Figure 4 (b). Please check.
-Line 356 - "rations” should be “ratios"
- Line 358 – “the peak near 1712 cm⁻¹ become more prominent” should be 'becomes more prominent".
- line 361/362 - “compounded through both SSE and TSSE indicates” should be “indicate”
- line 378 - “size nonuniformity” should be “size non-uniformity”
3.5. Rheological analysis of blend miscibility
- Passages like “this observation is concerned about the molten state…” or “a tail is located at the right end of the single arc” sound unnatural in technical English.
Additionally, there is redundancy in the use of “in both SSE and TSSE routes”.
- Avoid saying “Cole–Cole findings agree well with SEM morphological analysis” → use “The Cole–Cole analysis results are consistent with…” (more formal and precise).
3.6. Mechanical performance
- The text states that TSSE0 had a greater molecular weight reduction (18.4%) than SSE0 (1.5%), but this “does not directly correlate with the decrease in tensile strength”.
Although the decrease in strength is not always linear with molecular weight, the section could better explain why SSE0 maintains strength close to TSSE0 despite such a large difference in degradation. A more robust explanation would involve discussing the molecular weight distribution, polydispersity, and chain orientation during extrusion. Without this, the interpretation remains superficial.
- The text associates the reduction in tensile strength with the immiscibility between PLA and PBS and the low strength of PBS.
While immiscibility and PBS weakness partially explain this, the section ignores the possibility that domain size, morphology, and dispersion can strongly affect strength. For example, small, well-dispersed PBS inclusions could even locally improve toughness, but this is not discussed.
- The modulus consistently decreases with increasing PBS, which is expected. The text mentions that the slow decrease at low contents (5–15 wt.%) is due to partial miscibility and ultrafine PBS domain size, but it does not provide quantitative data on domain size or phase analysis (e.g., quantitative SEM or DMA). This makes the statement qualitative and unsubstantiated.
- EAB increases significantly with small additions of PBS and stabilizes above 15 wt.%. The text attributes this solely to 'morphological change and poor interfacial adhesion,' but it does not provide quantitative evidence of interfacial tension, phase separation energy, or detailed microstructure analysis. Without this, the conclusion is speculative.
- It is stated that both methods exhibit comparable mechanical behavior, but the data show differences: EAB and initial modulus (SSE0 vs TSSE0) are not identical. The molecular weight difference between SSE0 and TSSE0 is large, which should more clearly affect mechanical properties.
The conclusion of equivalence between SSE and TSSE is superficial, not taking into account the cumulative effects of PLA degradation.
- The section does not discuss extrusion speed, temperature, shear, and residence time, which can strongly influence modulus, strength, and EAB.
Attributing all changes solely to PBS content ignores critical processing variables.
- For example, sometimes it is written as “(Figure 6a, b)” and in other cases as “Figure 8 (a)”. Review and standardize throughout the text.
Conclusion
- It is stated that PLA underwent a greater molecular weight reduction in TSSE, but this did not result in significant differences in mechanical properties.
As discussed in Section 3.6, PLA degradation is 18.4% in TSSE versus 1.5% in SSE. It is unlikely that this would not partially impact tensile strength and modulus. A convincing explanation would require discussion of: Molecular weight distribution (polydispersity); Chain orientation during extrusion; Effect of submicron PBS domains on stress dissipation;
Without this, the conclusion of 'no significant difference' appears superficial.
- The conclusion states that up to 15 wt.% PBS, EAB increased to ~195% (vs 4.5% for processed PLA), with a moderate decrease in strength (~30%) and modulus (~15%).
The dramatic increase in EAB seems consistent with the data, but it is not discussed how fine PBS dispersion contributes mechanically.
The decline in strength and modulus could be elaborated in relation to interfacial tension and phase transition, which is not addressed.
- There is no mention of temperature, shear profile, or residence time, which are critical for shear-sensitive PLA and PBS.
This reduces the robustness of the claim that SSE is equivalent to TSSE.
Round 2
Reviewer 2 Report
Comments and Suggestions for Authors
The authors have thoroughly responded to all the comments and questionsI had.
They have made the proposed corrections.
Except on line 122 – “...of 1.26 g/cm³ and…”.
I also noticed that sometimes they write, for example, 608.0±17.3 and in other cases 608.0 ± 17.3. It would be better to standardize and include spaces.
Thus, it can be accepted.
